# KDM2A Deficiency in the Liver Promotes Abnormal Liver Function and Potential Liver Damage

**DOI:** 10.3390/biom13101457

**Published:** 2023-09-27

**Authors:** Matthew Martin, Aishat Motolani, Hyeong-Geug Kim, Amy M. Collins, Faranak Alipourgivi, Jiamin Jin, Han Wei, Barry A. Wood, Yao-Ying Ma, X. Charlie Dong, Raghavendra G. Mirmira, Tao Lu

**Affiliations:** 1Department of Pharmacology and Toxicology, Indiana University School of Medicine, 635 Barnhill Drive, Indianapolis, IN 46202, USA; marti336@gmail.com (M.M.); amotolan@iu.edu (A.M.); falipour@iu.edu (F.A.); jinjiamin@glmc.edu.cn (J.J.); xiaohan1119@gmail.com (H.W.); ym9@iu.edu (Y.-Y.M.); 2Department of Biochemistry and Molecular Biology, Indiana University School of Medicine, Indianapolis, IN 46202, USA; kim12@iu.edu (H.-G.K.); xcdong@iupui.edu (X.C.D.); 3Department of Pathology and Laboratory Medicine, Indiana University Health, Indianapolis, IN 46202, USA; acollins8@iuhealth.org (A.M.C.); bwood@iuhealth.org (B.A.W.); 4Indiana University Simon Comprehensive Cancer Center, Indiana University School of Medicine, Indianapolis, IN 46202, USA; 5Center for Diabetes and Metabolic Diseases, Indiana University School of Medicine, Indianapolis, IN 46202, USA; 6University of Chicago Medicine Kovler Diabetes Center, Chicago, IL 60637, USA; mirmira@uchicago.edu; 7Department of Medical & Molecular Genetics, Indiana University School of Medicine, Indianapolis, IN 46202, USA

**Keywords:** diabetes, KDM2A, liver, metabolism, NF-κB

## Abstract

Dysregulation of metabolic functions in the liver impacts the development of diabetes and metabolic disorders. Normal liver function can be compromised by increased inflammation via the activation of signaling such as nuclear factor (NF)-κB signaling. Notably, we have previously identified lysine demethylase 2A (KDM2A)—as a critical negative regulator of NF-κB. However, there are no studies demonstrating the effect of KDM2A on liver function. Here, we established a novel liver-specific *Kdm2a* knockout mouse model to evaluate KDM2A’s role in liver functions. An inducible hepatic deletion of *Kdm2a, Alb-Cre-Kdm2a^fl/fl^* (*Kdm2a* KO), was generated by crossing the *Kdm2a* floxed mice (*Kdm2a^fl/fl^*) we established with commercial albumin-Cre transgenic mice (B6.Cg-Tg(Alb-cre)21Mgn/J). We show that under a normal diet, *Kdm2a* KO mice exhibited increased serum alanine aminotransferase (ALT) activity, L-type triglycerides (TG) levels, and liver glycogen levels vs. WT (*Kdm2a^fl/fl^*) animals. These changes were further enhanced in *Kdm2a* liver KO mice in high-fat diet (HFD) conditions. We also observed a significant increase in NF-κB target gene expression in *Kdm2a* liver KO mice under HFD conditions. Similarly, the KO mice exhibited increased immune cell infiltration. Collectively, these data suggest liver-specific KDM2A deficiency may enhance inflammation in the liver, potentially through NF-κB activation, and lead to liver dysfunction. Our study also suggests that the established *Kdm2a^fl/fl^* mouse model may serve as a powerful tool for studying liver-related metabolic diseases.

## 1. Introduction

KDM2A is an important member of the JmjC domain histone demethylases. Traditionally, it catalyzes the demethylation of histone 3 lysine 36 (H3K36) and histone 3 lysine 4 (H3K4), thereby regulating important biological processes such as chromatin remodeling and cellular development [1,2]. In recent years, we and others discovered that KDM2A also modifies non-histone proteins, such as β-Catenin and NF-κB, to regulate their stability and transcriptional activity, respectively [3,4]. Given its extensive biological significance, KDM2A has been implicated in inflammatory-driven diseases, including cancer and obesity [1]. Thus, considering that liver function and obesity, diabetes [5], and other metabolic disorders are frequently interrelated, we aimed to assess the role of KDM2A in liver function.

Previously, we reported that KDM2A negatively regulates NF-κB signaling [4,6,7]. The NF-κB transcription factors are critical regulators of immune and inflammatory responses. They play key roles in chronic inflammation-related metabolic diseases including diabetes through the upregulation of tumor necrosis factor (TNF) α, resulting in liver necrosis and the regulation of TG accumulation and the development of fatty liver in the case of obesity [1,8,9,10,11,12]. Furthermore, diabetes databases indicate low KDM2A expression in pancreatic islets, adipose tissue, liver, skeletal muscle, and the hypothalamus [13].

Thus, in this study, we established a novel liver-specific *Kdm2a* KO mouse model to evaluate whether deficiency of KDM2A in the liver leads to increased expression of NF-κB target genes and abnormal liver function, which are characteristic of metabolic disorders and diabetes. An inducible hepatic specific *Kdm2a* KO was generated by crossing the *Kdm2a* floxed mice (*Kdm2a^fl/fl^*) we established with commercial albumin-Cre transgenic mice (B6.Cg-Tg(Alb-cre)21Mgn/J) (Figure 1A and Appendix A). We show that under a normal diet, *Kdm2a* KO mice exhibited increased serum ALT activity, L-type TG levels, and liver glycogen levels vs. WT animals. These changes were further enhanced in *Kdm2a* KO mice HFD conditions. We also observed a significant increase in NF-κB target gene expression in *Kdm2a* KO mice under HFD conditions. Similarly, the KO mice exhibited increased immune cell infiltration. Collectively, these data support the notion that liver-specific KDM2A deficiency may enhance inflammation in the liver, potentially through NF-κB activation, and lead to liver dysfunction, thus suggesting that KDM2A may be a promising therapeutic target for liver-related metabolic diseases. Moreover, this study also proves that the *Kdm2a^fl/fl^* mouse model we established could serve as a powerful tool to study the role of KDM2A in different tissue-specific pathophysiological conditions.

## 2. Materials and Methods

### 2.1. Mouse Model Generation and Determination

Mice were housed in the animal facility with controlled temperature (22 ± 2 °C), humidity (60 ± 5%), and regular 12:12 light/dark cycle. For the Alb-cre KDM2A model, we used an inducible hepatic deletion of *Kdm2a* (*Kdm2a* KO) by crossing *Kdm2a* floxed mice (*Kdm2a^fl/fl^,* referred to as wild-type) generated with the help of InGenious Target laboratory (Ronkonkoma, NY, USA) with albumin-Cre transgenic mice (B6.Cg-Tg(Alb-cre)21Mgn/J) from the Jackson Laboratory (Bar Harbor, ME, USA). NDEL2A and NDEL1A (NDEL2A and NDEL1A (NDEL2A 5′-CAAATAACAAGGCCTGGAGAGACGG-3′, NDEL1A 5′-CAATTT ACTCATACTTGGTGGTGTGCACC-3′) were used as targets for the screening of the Neo deletion in floxed mice by PCR of mouse tail or ear samples. Cre insertion was determined by albumin-cre-specific primers Cre-gt-42,5′-CTAGGCCACAGAATTGAAAGATCT-3′ Cre-gt-43 (5′-GTAGGTGGAAATTCTAGCATCATCC-3′) and Cre-gt-567 (5′-ACCAGCCAGCTATCAACTCG-3′) Cre-gt-568 (5′-TTACATTGGTCCAGCCACC-3′). Mice were also treated under a normal (6% energy by fat) or high-fat diet (HFD) (58% energy by fat, Research Diets Inc., New Brunswick, NJ, USA) for 4–5 weeks. All animal experiments performed were approved by IACUC (Institutional animal care and use committee).

### 2.2. Quantitative Polymerase Chain Reaction

Liver tissues were homogenized in Trizol. Then, the isolated RNA was used to prepare cDNA using the SuperScript III First-Strand Synthesis PCR System. qPCR was carried out using FastStart Universal SYBR Green Master ROX. Primers were designed by the NCBI Primer BLAST tool. The mouse primer sequences used were TNFα-Forward: 5′-GGTGCCTATGTCTCAGCCTCTT-3′; TNFα-Reverse: 5′-GCCATAGAACTGATGAGAGGGAG-3′; Vascular endothelial growth factor A (VEGFA)-Forward: 5′-CTTTCTGCTCTCTTGGGTGC-3′; VEGFA-Reverse: 5′-GCAGCCTGGGACCACTTG-3′; F4/80-Forward 5′-CCCAATGAGTAGGCTGGAGA; F4/80-Reverse: 5′-TCTGGACCCATTCCTTCTTG-3′; GAPDH-Forward: 5′-CCGGGTTCCTATAAATACGGACTG-3′; GAPDH-Reverse: 5′-CCAATACGGCCAAATCCGTT-3′.

### 2.3. Serum Biochemistry

Serum ALT and L-type TG were analyzed using commercial kits (Thermo Fisher, Wako, TX, USA) following the manufacturer’s manuals. Serum was measured in both normal and HFD animals under a 4-week HFD.

### 2.4. Histological Analysis

Liver tissue samples under normal or HFD conditions were fixed in 10% formalin and then processed for embedding and sectioning at the Histology Core of Indiana University School of Medicine. Liver sections (5 μm thickness) were stained using H & E following the standard protocol. Immunofluorescence (IF) analysis was performed for detecting inflammation by inflammatory cell markers including F4/80 (Thermo Fisher, MA5-16363) and desmin (DSHB, D3) and detecting fibrosis by Sirius red (Millipore Sigma, St. Louis, MO, USA) staining analyzed by immunohistochemistry (IHC) via light microscopy. Images for H & E and IHC were captured using a Leica DM750 microscope equipped with an EC3 digital camera and LAS EZ software. IF images were obtained by a ZEISS fluorescence microscope or a Leica DMI6000B microscope. Images were analyzed using Image J 1.44.

### 2.5. Liver Glycogen Measurement

To measure the liver glycogen content, Periodic Acid Schiff (PAS) staining was performed. Paraffin-embedded livers were sectioned at 5 μm, deparaffinized, and then stained with Periodic Acid and Schiff’s reagent. Slides were further stained with hematoxylin solution to visualize them. Imaging was performed using the Leica DM750 microscope equipped with an EC3 digital camera and LAS EZ software. The nucleus is stained in blue-purple while glycogen is stained in magenta. Quantifications were performed using the color deconvolution tool on ImageJ 1.44.

### 2.6. Fasting/Non-Fasting Blood Glucose

Mice were either fasted, with free access to water, for 16 h overnight or non-fasted at the same time of day in the morning. Blood glucose was measured in the tail vein using contour blood glucose test strips with a Contour Blood Glucose Monitoring System (Fisher, 22-021-515, Waltham, MA, USA). Blood was drawn from mice at 8–9 weeks of age. Blood was drawn from mice under normal or 4-week HFD conditions.

### 2.7. Statistical Methods

The data represent the means ± SD from at least three separate experiments performed in triplicate. The differences between groups were analyzed using a two-tailed Mann–Whitney U test. A *p*-value of <0.05 was considered statistically significant. Statistical analyses were carried out using GraphPad Prism (9.0).

## 3. Results

Recently, the whole-body *Kdm2a* KO mice model was reported to exhibit embryonic lethality at E10.5–12.5l, limiting its applications [8]. Thus, to study the role of KDM2A in the liver, we established a conditional *Kdm2a* KO mouse model (Figure 1A and Appendix A). Briefly, liver-specific *Kdm2a* KO mice were generated by crossing *Kdm2a flox* (*Kdm2a^fl/fl^*) mice (Figure 1A) with liver-specific albumin-Cre transgenic mice (B6.Cg-Tg(Alb-cre)21Mgn/J). The design of the floxed region in *Kdm2a* is aimed at the region between exons 9 and 10, which is the crucial region for the KDM2A enzymatic activity. Using a PCR genotyping approach, we further confirmed the correct genotypes of *Kdm2a* KO or WT using mouse tissue. As shown in Appendix A, *Kdm2a* liver KO and WT mouse samples were screened and used for experiments. Both female and male mice (age: 8–9 weeks) were used. First, to assess liver function, we wondered whether *Kdm2a* KO in the liver could alter blood glucose regulation. Therefore, we determined blood glucose levels under both non-fasting and fasting conditions with a normal diet. As shown in Figure 1B, we did not observe significant differences in blood glucose levels between WT and KO in both non-fasting and fasting states. Additionally, since both ALT and TG levels are commonly used benchmarks for liver dysfunction in metabolic disorders and may indicate liver damage [10,11,13], we further measured the levels of ALT and TG in WT and *Kdm2a* liver KO mouse serum. As shown in Figure 1C, ALT (left panel) showed an increased trend (though not significant) in *Kdm2a* liver KO mice as compared to the WT mice. Interestingly, when analyzed within each gender, both male and female mice showed significantly increased ALT activity in *Kdm2a* liver KO as compared to WT mice (Appendix A). Moreover, we observed that TG levels (Figure 1C, right panel) were significantly higher in *Kdm2a* liver KO mice in comparison to the WT mice. Further analysis within each gender also showed markedly higher levels of TG in *Kdm2a* liver KO mice (Appendix A). Together, these data indicate potential damage in *Kdm2a* KO livers.

The storage ability of glycogen in the liver is another perspective for measuring liver function. To measure the liver glycogen content, PAS staining was performed. Slides were further stained with hematoxylin solution (Figure 1D). Data suggest that glycogen levels were significantly higher in *Kdm2a* liver KO mice in comparison to the WT mice, suggesting higher glycogen storage in *Kdm2a* KO mouse livers. Moreover, we further conducted H & E staining to determine if there were clear morphological differences between WT and *Kdm2a* KO mouse livers. As shown in Figure 1E, no significant morphological differences were observed between the livers of *Kdm2a* KO and WT mice under a normal diet, per the pathologist’s verification.

Obesity is a major hallmark for the development of insulin resistance and results in liver dysfunction [10,11]. We performed an HFD challenge with mice aged 8–9 weeks to determine if an HFD exacerbated potential liver damage and reduced liver function. As shown in Figure 2A, under HFD conditions, *Kdm2a* liver KO mice exhibited dramatically increased fasting blood glucose levels compared to WT mice. A similar trend of increased non-fasting blood glucose levels was also observed in *Kdm2a* liver KO as compared to WT animals though was not statistically significant. Similar to what we have tested in normal diet conditions, we also determined the levels of ALT and TG under HFD conditions. As shown in Figure 2B, *Kdm2a* liver KO animals had a marked increase in the serum levels of ALT (left panel) and TG (right panel) in comparison to the WT animals, which were higher than the corresponding levels obtained under normal diet conditions (Figure 1C). To evaluate the influence of the types of diet on *Kdm2a* liver KO mice, we compared levels of blood glucose, ALT, and TG in the KO mice under a normal diet and an HFD. Herein, we observed that the HFD significantly increased blood glucose and TG levels, with an insignificant increase in mean ALT levels (Figure 2C). We also conducted PAS staining to determine the liver glycogen content in these HFD-fed mice (Figure 2D). Data suggest that glycogen levels were significantly higher in *Kdm2a* liver KO mice in comparison to the WT mice, indicating a markedly higher glycogen content in *Kdm2a* KO mouse livers. To determine if the HFD increased potential liver damage and dysfunction in *Kdm2a* liver KO animals, we determined the morphology of animal livers by H & E staining (Figure 2E). No significant morphology changes were observed. We then explored potential inflammatory marker differences potentially dysregulated by *Kdm2a* liver KO by staining HFD livers with Sirius red staining to evaluate the collagen proportional area (CPA) as a marker for fibrosis. As shown in Figure 3A, fibrosis was increased in *Kdm2a* liver KO mice compared to WT mice. It is well known that high TNFα and VEGF levels have been linked to an increased risk of liver-related dysfunction, such as nonalcoholic fatty liver diseases, liver cirrhosis, apoptosis, etc. [14,15]. Importantly, these two inflammatory factors are also prototypical NF-κB target genes that may be negatively regulated by *Kdm2a*. Thus, we assessed the transcript levels of TNFα and VEGF. In Figure 3B, we observed that *Kdm2a* liver KO mice showed significantly increased expression of TNFα (Top panels) and VEGFA (Bottom panels) under HFD conditions. A similar trend was observed in mice on a normal diet (Figure 3B). These results not only proved the increased expression of cytokines in the *Kdm2a* liver KO mice as compared to the WT mice but also demonstrated the activation of NF-κB, as both TNFα and VEGF are well-known target genes of NF-κB. This is consistent with our previous discovery that KDM2A is a negative regulator of NF-κB [4,6]; thus, *Kdm2a* liver KO led to the activation of NF-κB.

Given the importance of hepatic stellate cells (HSCs) in liver inflammation, we stained for desmin, a marker of HSCs. HSCs are commonly considered the precursor population of septal myofibroblasts (MF) in fibrosis and cirrhosis [16]. Our data showed that the number of HSCs increased in the KO livers (Figure 3C, top panels). We further stained macrophage expression with F4/80 (green) to determine differences in immune cell infiltration in mouse livers. As shown in Figure 3C (Top panels), *Kdm2a* KO mouse livers exhibited higher macrophage infiltration around both the portal and hepatic veins. To confirm this phenomenon, we further conducted qPCR experiments for the expression of F4/80. As shown in Figure 3C (Bottom panel), there is a significant increase in F4/80 transcripts in *Kdm2a* liver KO mice compared to the WT. Together, these results suggest the critical role of *Kdm2a* in regulating the expression of inflammatory markers that contribute to liver damage.

## 4. Discussion

KDM2A has been studied in cancer and inflammatory diseases [1,4,6,7,8]. Despite KDM2A’s clear role in the regulation of NF-κB signaling pioneered by our group [4,6], there is minimal information regarding its functions in the liver. Our work here explores KDM2A’s role in hepatic metabolic diseases and disorders by using a liver-specific *Kdm2a* KO mouse model generated by us with the help of the InGenious Target laboratory, Ronkonkoma, NY. Overall, we show that our *Kdm2a* liver KO mice exhibited increased serum ALT activity, TG levels, and liver glycogen levels vs. WT animals. These changes were further enhanced in *Kdm2a* liver KO mice under HFD conditions. Notably, nonalcoholic fatty liver disease (NAFLD) is a frequent occurrence in liver-related metabolic diseases such as obesity and type 2 diabetes mellitus (T2DM) [17]. The incidence of metabolic syndrome, as characterized by increased ALT and TG, has been reported to be significantly higher in NAFLD patients compared to non-NAFLD patients [18]. Given our findings, it is reasonable to speculate that *Kdm2a* liver deficiency may contribute to the incidence of high metabolic syndrome in NAFLD patients. This would be an interesting aspect warranting further investigation in the future.

The regulation of glucose levels by the liver is critical to fueling the energy demands of cells. During fasting, the liver stimulates the production of glucose through glycogenolysis as a way of maintaining blood glucose levels. This glucose production process is increasingly exacerbated in liver diseases such as T2DM, thus leading to hyperglycemia [19]. Notably, hyperglycemia is a major hallmark of diabetes. *Kdm2a* has been reported to negatively regulate the expression of gluconeogenic genes in hepatocytes and mice and, thus, speculated to protect against diseases like Type 2 diabetes [20]. Moreover, Pan and colleagues have demonstrated that the lentiviral knockdown of *Kdm2a* increases hepatic gluconeogenesis via its reduced dimethylation activity on the C/EBPα promoter region [21]. However, they emphasized that the role of *Kdm2a* in gluconeogenesis in the liver specific *Kdm2a KO* model remains to be addressed and elucidated. In this study, using our novel *Kdm2a* liver-specific KO model, we uncovered the critical role of KDM2A in the regulation of fasting blood glucose levels under HFD conditions. Considering that a previous study reported that hyperglycemia is induced in mice within 4 weeks of an HFD [22], the significant difference observed in blood glucose levels under an HFD (Figure 2A) compared to a normal diet (Figure 1B) between WT and KO fasting mice is unsurprising. More studies will be required to elucidate the genes mediating glucose regulation under HFD downstream of *Kdm2a*. Additionally, our study suggests that *Kdm2a* is crucial for liver glycogen levels. This could result from increased glucose synthesis mediated by *Kdm2a* silencing, as reported by Chen et al. [20], a speculation that merits further research in the future. Collectively, our study has provided strong evidence regarding the critical role of *Kdm2a* in liver homeostasis.

Furthermore, NF-κB-dependent mechanisms have been reported to play a major role in the hepatic disease process, including fibrosis and chronic inflammation. Moreover, NF-κB has also been linked to the survival and activation of HSCs, leading to increased cytokine production and liver fibrosis [23]. Our data in Figure 3 suggest the potential contribution of proinflammatory cytokines genes such as *TNFα* and *VEGFA* to liver dysfunction via KDM2A. Similarly, the increased macrophage infiltration induced by *Kdm2a* KO in the liver may be driven by the aforementioned cytokines, among others. Karlmark and colleagues have reported that different activated hepatic cell populations secrete cytokines, which direct the infiltration of immune cells [24]. This phenomenon is a critical aspect of acute and chronic liver diseases. Further studies will be required to assess the role of KDM2A in recruiting other types of immune cells in a cytokine-dependent manner in various liver diseases.

Moreover, our novel conditional *Kdm2a* liver KO mouse model is proven to be an effective and more ideal model than the whole-body *Kdm2a* KO mouse model, as the latter is embryonically lethal [8]. This new conditional *Kdm2a* liver KO mouse model opens interesting avenues to explore in the future, such as determining KDM2A’s effect on hepatocytes, or mechanically inducing liver damage with acetaminophen. Additionally, pursuing longer HFD experiments will determine if later stages of obesity exacerbate damage in *Kdm2a* liver KO mice. Collectively, our data have demonstrated the novel role of KDM2A in liver function, suggesting its potential as a promising therapeutic target in inflammatory metabolic disorders. Our novel conditional *Kdm2a^fl/fl^* mice can be bred with other tissue specific Cre mice. Thus, the resulting model can be used to study other tissue-specific disorders that may be related to KDM2A deficiency.

## Figures and Tables

**Figure 1 biomolecules-13-01457-f001:**
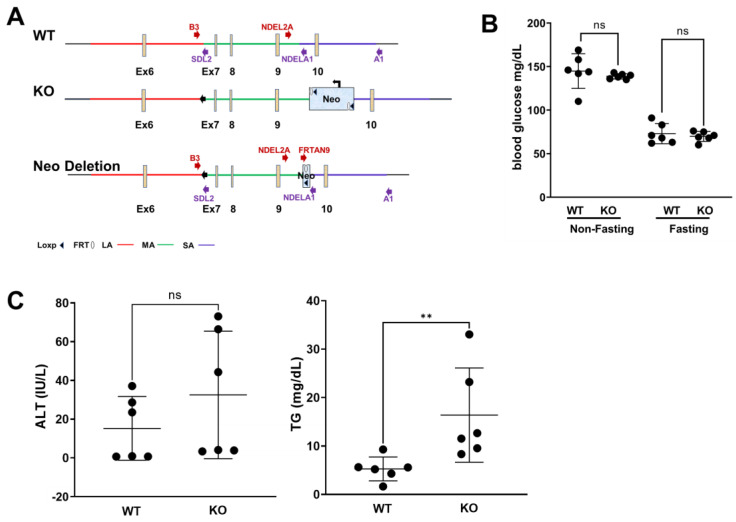
*Kdm2a* liver KO mice exhibit a disease-like state in liver under normal diet compared to WT mice. (**A**) Schematic illustration of the design and detection strategy of *Kdm2a^fl/fl^* gene (referred to as WT), KO, and Neo deletion. Targeted iTL IC1 (C57BL/6) embryonic stem cells were microinjected into Balb/c blastocysts and the resulting chimeras with a high percentage black coat color were mated to C57BL/6 FLP mice to remove the Neo cassette. For additional method details, including primer sequences, please refer to Method section—“Mouse model generation and determination”. (**B**) Blood glucose is similar in WT and *Kdm2a* liver KO mice under normal diet. Fasting and non-fasting blood glucose showed no significant differences between *Kdm2a* liver KO and WT mice. N = 6. (**C**) *Kdm2a* liver KO mice exhibited increased levels of serum ALT (n.s.) and significantly increased L-type TG as compared to WT animals. N = 6, ns = non-significant, ** *p* < 0.002, WT vs. *Kdm2a* liver KO TG. (**D**) Left panels: Representative images of Periodic Acid and Schiff’s (PAS) reagent staining on WT and *Kdm2a* liver KO liver sections at 20× and 40× magnification. The nucleus is stained blue, and glycogen is stained magenta. Bottom panel: ImageJ quantification of three different sections of each liver sample, indicating significant increase in liver glycogen in *Kdm2a* liver KO mice on normal diet. N = 3. Mann–Whitney U test * *p* < 0.05. (**E**) H & E staining of mouse livers, indicating no significant difference in liver morphology between *Kdm2a* liver KO and WT mice under a normal diet.

**Figure 2 biomolecules-13-01457-f002:**
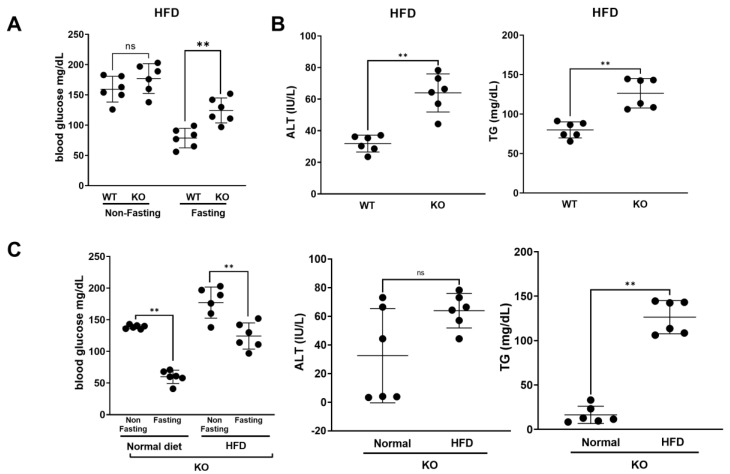
*Kdm2a* liver KO mice exhibit a disease-like state in liver under HFD conditions compared to WT mice. (**A**) *Kdm2a* liver KO mice exhibited increased levels of fasting blood glucose as compared to WT mice under HFD conditions. N = 6. Mann–Whitney U test: ns = non-signifcant, ** *p*-value < 0.002, *Kdm2a* liver KO fasting vs. WT fasting. (**B**) *Kdm2a* liver KO animals exhibited significantly increased levels of serum ALT and serum L-type TG as compared to WT mice under HFD conditions. N = 6, Mann–Whitney U test: ** *p* < 0.002, WT vs. KO ALT, WT vs. KO TG. (**C**) Comparison of levels of blood glucose, ALT, and TG in *Kdm2a* liver KO mice under normal and HFD conditions. Mann–Whitney U test: ** *p*-value < 0.002 (**D**) Top panels: Representative images of PAS reagent staining on *Kdm2a* WT and KO liver sections at 20× and 40× magnification. The nucleus is stained blue and glycogen is stained magenta. Bottom panel: ImageJ quantification of three different sections of each liver sample, indicating significant increase in liver glycogen in *Kdm2a* liver KO mice on an HFD. N = 3. Mann–Whitney U test * *p* < 0.05. (**E**) H & E staining of mouse livers, indicating no significant difference in liver morphology between *Kdm2a* liver KO and WT mice under HFD conditions.

**Figure 3 biomolecules-13-01457-f003:**
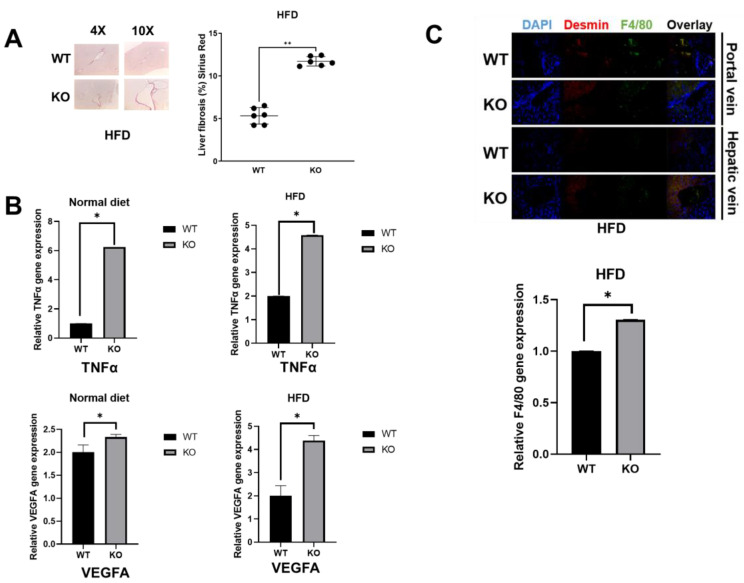
*Kdm2a* liver KO mice exhibit increased fibrosis, proinflammatory cytokine expression, and macrophage expression in liver under HFD conditions. (**A**) Sirius Red staining of fibrosis of livers was increased in *Kdm2a* liver KO animals compared to WT animals under HFD conditions. Staining of fibrosis via Sirius red staining was quantified in ImageJ. N = 6, Mann–Whitney U test ** *p* < 0.002, *Kdm2a* liver KO 6 × livers vs. WT 6 × livers. (**B**) qPCR analysis, showing increased expression of NF-κB target cytokine genes, TNFα and VEGFA, in *Kdm2a* liver KO as compared to WT animals. N = 4 * *p* < 0.05. (**C**) Top panels: IHC staining, showing that *Kdm2a* liver KO animals exhibited increased macrophage and HSC expression in mouse livers compared to WT animals under HFD conditions. Desmin is used as an HSC marker. N = 3. Bottom panel: qPCR analysis, showing significant increase in F4/80 transcript levels in *Kdm2a* KO liver compared to the WT. F4/80 is a macrophage marker. N = 4 * *p* < 0.05.

## Data Availability

Data available upon request.

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
