# Peer review of "KDM2A Deficiency in the Liver Promotes Abnormal Liver Function and Potential Liver Damage"

_biomolecules, 2023, doi:10.3390/biom13101457_

Round 1

Reviewer 1 Report

In this manuscript, the authors have developed a novel liver-specific Kdm2a knockout mouse model to elucidate the role of KDM2A in liver functions. They found that under standard diet conditions, Kdm2a KO mice displayed an elevation in serum alanine aminotransferase (ALT) activity, heightened levels of L-type triglycerides (TG), and increased liver glycogen levels compared to their wild-type mice. Additionally, metabolic alterations were further exacerbated when Kdm2a KO mice were subjected to a high-fat diet (HFD). Furthermore, this study demonstrated a significant upregulation in the expression of NF-κB target genes in the liver tissue of Kdm2a KO mice under HFD conditions. Concomitantly, these knockout mice showed an augmented infiltration of immune cells into the liver parenchyma. These data suggest that the deficiency of KDM2A specifically within the liver microenvironment may potentiate hepatic inflammation, likely through the activation of NF-κB signaling pathways, ultimately leading to impaired liver function.

Minor comments:

As shown in Fig 1, TG levels were increased in KO mice compared to WT mice. What about the levels of cholesterol or low-density lipoproteins under normal diet conditions?

In Fig 2, please include a comparison between normal diet and HFD for glucose, ALT, and TG levels in KO mice.

Author Response

Dear Reviewer,

We would like to sincerely thank you for reviewing our manuscript entitled “KDM2A deficiency in the liver promotes abnormal liver function and potential liver damage” and taking it under consideration for publication. Below, we have fully addressed your comments point-by-point.

Point 1: As shown in Fig 1, TG levels were increased in KO mice compared to WT mice. What about the levels of cholesterol or low-density lipoproteins under normal diet conditions?

Response 1: This is a very insightful question. In our study, we only measured TG levels as a representative indication of lipid metabolism. Given the critical role of KDM2A in liver metabolism that we have proven in this study, it would be very interesting to conduct Lipidomic analysis in both WT and KDM2A liver KO mice in the future. This would provide us with a complete profile of eight categories of lipids, including fatty acyls (FA), glycerolipids (GL), glycerophospholipids (GP), sphingolipids (SP), sterol lipids (ST), prenol lipids (PR), saccharolipids (SL), and polyketides (PK) in these two different mouse models, thus deepening our understanding of the role of KDM2A in the overall lipid metabolism.

Point 2: In Fig 2, please include a comparison between normal diet and HFD for glucose, ALT, and TG levels in KO mice.

Response 2: Thanks for your very valuable suggestion. We have included the comparison stated in the new Figure 2C. Corresponding text description has also been incorporated in both main text and the revised figure legend.

Reviewer 2 Report

In the paper entitled “KDM2A deficiency in the liver promotes abnormal liver function and potential liver damage” Martin and co-workers focused their efforts in understanding the role of lysine demethylase 2A (KDM2A) in liver damage; Authors generated KDM2A knockout mice that carried out this mutation only in liver, since the KDM2A whole body mutation does not allow to obtain living mice.  Martin and colleagues concluded that liver-specific KDM2A deficiency may enhance inflammation in the liver suggesting KDM2A could represent a potential target in inflammation mediated liver disease.

The paper is well written, data flow is consistent and results are clearly presented; rationale is clear and both figures and figure legend are exhaustive. 

Nevertheless, some concerns are to be assessed to the Authors:

1-     Discussion section is very poor. This section needs to be definitely rewritten since it’s not actually a discussion according to scientific standards: Authors summarized their result without any explanation nor data justification. Authors are strongly encouraged to re-write this section.

2-      Scientific soundness is not clear as well as Author’s findings translational significance; Author referred in the paper about their findings as potentially powerful tool to study tissue specific disease without explain which disease are they talking about: this point deserves more and deep elucidation

3-     Statistical methods need to be deeply elucidated. Authors better state which statistical test they used for their analysis. I suggest using Mann-Whitney t test. In any case, statistical method explanation need to be improved.

Author Response

Dear Reviewer,

We would like to sincerely thank you for reviewing our manuscript entitled “KDM2A deficiency in the liver promotes abnormal liver function and potential liver damage” and taking it under consideration for publication. Below, we have fully addressed your comments point-by-point.

Point 1: Discussion section is very poor. This section needs to be definitely rewritten since it’s not actually a discussion according to scientific standards: Authors summarized their result without any explanation nor data justification. Authors are strongly encouraged to re-write this section.

Response 1: Thanks very much for this critical suggestion. We have rewritten the discussion section with more context and data justification.

Point 2: Scientific soundness is not clear as well as Author’s findings translational significance; Author referred in the paper about their findings as potentially powerful tool to study tissue specific disease without explain which disease are they talking about: this point deserves more and deep elucidation.

Response 2: Thanks for your helpful suggestion. We have rephrased the sentences to be more specific.

Point 3: Statistical methods need to be deeply elucidated. Authors better state which statistical test they used for their analysis. I suggest using Mann-Whitney t test. In any case, statistical method explanation need to be improved.

Response 3: Thanks for your comment. We have changed the statistical test performed to the Mann-Whitney U test, using the Prism 9.0 software. This has been stated in both the Methods section and Figure Legend section.

We thank you for your helpful feedback and suggestions. We hope you will agree with our effort and find this manuscript acceptable for publication.

Round 2

Reviewer 2 Report

In revised version of the paper entitled “KDM2A deficiency in the liver promotes abnormal liver function and potential liver damage” Martin and co-workers improved the discussion and soundness and elucidated statistical methods applyng all the suggestions requested.